# Social Deprivation, Healthcare Access and Diabetic Foot Ulcer: A Narrative Review

**DOI:** 10.3390/jcm11185431

**Published:** 2022-09-15

**Authors:** Jean-Baptiste Bonnet, Ariane Sultan

**Affiliations:** 1Nutrition-Diabetes Department, University Hospital of Montpellier, 34295 Montpellier, France; 2UMR 1302, Desbrest Institute of Epidemiology and Public Health, INSERM, CHU, University of Montpellier, 34000 Montpellier, France; 3PhyMedExp, INSERM U1046, CNRS UMR 9214, University of Montpellier, 34000 Montpellier, France

**Keywords:** diabetic foot ulcer, epidemiology, social deprivation

## Abstract

The diabetic foot ulcer (DFU) is a common and serious complication of diabetes. There is also a strong relationship between the environment of the person living with a DFU and the prognosis of the wound. Financial insecurity seems to have a major impact, but this effect can be moderated by social protection systems. Socioeconomic and socio-educational deprivations seem to have a more complex relationship with DFU risk and prognosis. The area of residence is a common scale of analysis for DFU as it highlights the effect of access to care. Yet it is important to understand other levels of analysis because some may lead to over-interpretation of the dynamics between social deprivation and DFU. Social deprivation and DFU are both complex and multifactorial notions. Thus, the strength and characteristics of the correlation between the risk and prognosis of DFU and social deprivation greatly depend not only on the way social deprivation is calculated, but also on the way questions about the social deprivation−DFU relationship are framed. This review examines this complex relationship between DFU and social deprivation at the individual level by considering the social context in which the person lives and his or her access to healthcare.

## 1. Introduction

Social deprivation is defined as “the inability of individuals to participate fully in the life of their community or society” [1]. This notion is more focused on “material or financial resources”, whereas social exclusion tends to emphasize “the lack of participation in a broader range of social, cultural, and political activities” [1]. More specifically, the notion of social exclusion is of particular interest for healthcare actors as it encompasses the issue of access to the healthcare system itself. The literature on the diabetic foot ulcer (DFU) uses the term “social deprivation” in a broad sense, which leads us to differentiate between strict financial deprivation and social deprivation in the same broad sense, thus including social exclusion [2]. While most studies on this topic have focused on a hierarchy based on household income or the type of health insurance, several healthcare systems have seen the emergence of more refined measurement methods. Examples include the individual EPICES score [3] or methods that assess the person within the context of a complex living environment, as may be done in England or Scotland [2].

Lower limb trophic disorder in people living with diabetes, often referred to as “diabetic foot”, is a common and serious complication of diabetes. It is estimated that a person living with diabetes has a 19–34% lifetime risk of developing a DFU [4]. This wound often becomes infected and in 20% of the cases the infection will lead to amputation [5], making diabetes the leading cause of non-traumatic amputation in Western countries [6]. Thus, DFU has a major social impact and, as one of the most common complications of diabetes, it requires the highest level of care. Its management relies on multidisciplinary teams, home care with often daily nursing and dressings, and respect for Off-loading, a key element of prognosis. Indeed, DFU has become a very serious public health issue. In addition to its social impact, the overall annual cost of managing DFU is very high, estimated in the United States alone at over 17 billion dollars [7].

As the amputation rate in a population is a significant marker of the quality of the healthcare system [8], it is monitored by international organizations such as the Organization for Economic Cooperation and Development (OECD). Links have been established between diabetes complications and social deprivation [9,10]; glycemic control and social deprivation [11]; diabetes, cardiovascular risk factors and social deprivation [12]; mortality, diabetes and social deprivation [13]; and diabetes incidence and social deprivation [14,15]. However, the literature is less extensive on the diabetic foot and its complications. It is therefore crucial to understand the links between this pathology, the healthcare system and socioeconomic characteristics.

We notably hypothesize that this complication of diabetes is particularly influenced—for both risk and prognosis—by the patient’s environment, especially their social status, as it might well impact healthcare needs. We further hypothesize that access to care also influences both the risk and prognosis of DFU. This can be measured in several ways: access to an attending physician (distance or time), to a nurse or, more specifically, for a pathology that requires many specialists, to a specialized multidisciplinary team and their equipment (e.g., endovascular surgery). Yet, data on the prevalence or incidence of DFU are missing in many databases. For this reason, it is often necessary to focus on lower limb amputations with the attendant risks of bias (not including only DFU). In this review, we systematically differentiate between the risk of DFU incidence (when the database studied offered this data item) and the risk of amputation/mortality (DFU prognosis) when this is the only data available. Time to healing is unfortunately difficult to evaluate, which explains why it often absent from studies.

The objective of this review is to provide an overview of the literature on the relationship between social deprivation and DFU (both risk and prognosis). This will be analyzed at three levels: the individual patient, the neighborhoods that patients live in, and the access to healthcare. Our objectives are to answer the following questions:-Is there a relationship between individual social deprivation markers and both DFU risk and prognosis?-Is there a relationship between neighborhood social deprivation markers and both DFU risk and prognosis?-Is there a relationship between healthcare access markers and both DFU risk and prognosis?

## 2. Method

We conducted a systematic search in the MEDLINE database. We focused on “original articles” and used the words “diabetic foot ulcer”, “diabetes”, “amputation”, “deprivation”, “ethnic”, and “socioeconomic”. The relevance of an article to the topic was assessed by its title and summary, followed by a full reading of the text; relevant articles were then integrated into the review. The selected article references were systematically examined in order to broaden the search. All selected articles had to be in English or French.

## 3. Relationship between Individual Social Deprivation Markers and Both Risk and Prognosis of DFU

### 3.1. Financial Deprivation

In 2016, *Santé Publique France* conducted a large study on the household income burden of diabetes and its complications in the country. The most socially deprived individuals under 60 years of age, who benefited from Complementary Universal Medical Coverage (CMU-C), had a 1.4 greater risk of being hospitalized because of DFU and a 1.7 greater risk of lower limb amputation than the general population. According to the regulations in place, individuals lose CMU-C at 60 years and it is replaced by other, less well-monitored health insurance systems. The 2016 study had the merit of using a “hard” criterion based on average income, despite this being a yes or no social indicator below a certain income level [16]. The large French prospective survey ENTRED reported similar data. Indeed, the analysis of nearly 4000 people living with type 2 diabetes in 2007 showed a prevalence of DFU inversely proportional to socio-educational level and a relative risk of 1.7 of DFU (CI95%; 1.4–2.2) for those reporting financial difficulties [17]. An update of the ENTRED study is currently in progress.

It is interesting to note that in two American states, Weissman et al., found higher rates [2.27 (CI95% 1.57–2.98) and 1.53 (CI95% 1.18–1.89) relative risk] of hospitalization for gangrene (person with diabetes and person without diabetes) for those without insurance or with Medicaid insurance than for those with private insurance. There was even a tendency toward over-risk for the subjects with Medicaid coverage, which is a lower quality of social coverage than Medicare. The lack of protection for these hospitalizations seems to indicate that the cost of care is not the only epidemiological factor of poor DFU evolution [18], but that household income is independently related to the risk of DFU occurrence and its poor prognosis.

The diabetic foot requires complex management that can have a number of hidden social and medical costs. It is interesting to note that in a country like France where healthcare is provided free of charge, shoes for transitory offloading still cost on the order of 30 to 50€ per shoe. In many insurance systems, the costs of transport, negative pressure therapy, certain dressing techniques, etc. must be added in. The impact of these residual charges on the adherence to treatment of individuals with a DFU remains inadequately studied [19].

### 3.2. Going beyond Financial Deprivation to Examine Social Deprivation and More Complex Socio-Educational Factors

Social deprivation is another aspect of deprivation that needs to be evaluated. The EPICES questionnaire was constructed to highlight forms of social deprivation not detected by socio-administrative criteria [3]. It gives a score that takes into account not only income, but also the social deprivation that individuals may experience, particularly regarding their ability to call on their family to help them. In a population of people living with diabetes, Bihan et al., showed an inverse relationship between their glycemic control and social deprivation. However, neither DFU [20] nor neuropathy, which is the main component of DFU, appears to be statistically over-represented in the most precarious group [9]. The quality of life in general nevertheless stands out as the first to be impacted by a lower EPICES score [20].

The relationship that links the patient to the pathology being studied may thus be more complex than a simple economic relationship. Socio-educational level and professional status certainly have an effect on household income, but they can act independently on health [21] and, for diabetes, on mortality [22] or micro-angiopathic complications [23]. For example, men’s health status, unlike that of women, is more affected by education and occupation than by household income [21], particularly in type 2 diabetes [24]. Women’s health seems to be impacted mainly by their “social class” [23]. This is in combination with the observation that women appear to be less well-monitored than men for DFU risk [25]. Therefore, the level of education appears to play a role in regular foot monitoring [26].

In England, Bachmann et al., had an interesting approach in their analysis of the Somerset and Avon Survey of Health (SASH) cohort. They compared general practitioner reports of people living with diabetes with the results of self-questionnaires administered to the same patients. Regarding other complications of diabetes such as diabetic retinopathy, there was a statistically significant inverse relationship between people’s incomes and neuropathy in the self-questionnaire, but not in their physicians’ reports. This difference was not observed by performing the same analysis with socio-educational level [27]. This finding confirms not only the difference between financial deprivation and socio-educational level, but also the necessity of dealing with the results of retrospective studies that have analyzed a single clinical criterion such as diabetic neuropathy, which has often been misclassified or poorly explored in the most socially disadvantaged populations. The analysis of cohorts must take into account the social reality of the population under study. Koskinen et al., noted that their Finnish studies showed much less marked results than those from the UK [28,29] as the Finnish population has fewer social inequalities than the British population. Thus, in a population with fewer income or socio-educational inequalities, fewer differences will be observed [28].

Further, adherence to treatment is critically important. In the same study of the SASH cohort, Bachmann et al., showed that, although physicians were more likely to see people with low incomes or low levels of education as being less compliant with care than those with higher education levels or high incomes, this was not the case for the self-administered surveys. This points to a mismatch between the compliance perceived by the physician and that perceived by the patient. On the other hand, people with low incomes reported more negative life events from diabetes and were significantly more afraid of diabetes and its complications [27]. The difference between the self-perceived state of health and that reported by the medical staff based on income or socio-educational level is not specific to diabetes [21].

Regarding the ethnic factor, the New Zealand cohort study by Gurney et al. [30] highlighted an association between ethnic origin and the risk of amputation in people with diabetes. Indeed, people of Māori origin, as Afro-Americans in the United States, were found to be statistically more disadvantaged than Caucasian populations, and people of Asian origin were not always disadvantaged.

## 4. Relationship between Neighborhood Social Deprivation Markers and Both Risk and Prognosis of DFU

In many retrospective studies, the socioeconomic level of the patients is unknown but their place of residence is noted. Therefore, we have data analyzing the relationship between the socioeconomic level of the place of residence and the prevalence of DFU and its prognosis. This analysis is very much relevant for a pathology such as DFU, which requires a high level of care. The management of this care is necessarily part of the health policy within a given territory and depends on the care supply for that territory. On an international level, the rate of amputations among people with diabetes can vary by a factor of 20 across the OECD countries [8].

### 4.1. Analysis on a Regional or Municipal Scale

Several studies have explored small-scale habitat areas. One was conducted in California and focused on Los Angeles [31], another was conducted in metropolitan Glasgow, Scotland [2], and another in the largely rural region of Cheshire, England [32]. The California study showed a linear relationship between the average per capita income in the residential area and the incidence of amputations. The Glasgow study had the advantage of using the Scottish government’s complex scores for monitoring the social deprivation of neighborhoods defined by a population of 600. The Scottish Index of Multiple Deprivation (SIMD) combines analyses of average income, education, unemployment, crime, access to public services, and so on. Hurst et al.’s analysis [2] showed a major threshold effect within the poorest quintile of neighborhoods on the incidence of DFU, amputations and mortality. The protective effect of the most advantaged neighborhoods was much less clear-cut: it was most evident for the risk of amputation and much less so for the risk of DFU or post-DFU mortality. According to the SIMD, the threshold effect of the poorest quintile of neighborhoods on the risk of DFU had already been found in Scotland in the Tayside agglomeration, giving more weight to this Scottish observation [32]. In the county of Cheshire, England, Anderson et al., used a less developed tool than the SIMD, the Townsend index, which focuses on the unemployment rate, the proportion of people without a car, and the quality of housing [32,33]. This more linear link was also observed in the California study [31], which included only an analysis of average neighborhood incomes. In the English study, the impact of a neighborhood’s social deprivation had even greater statistical power than age, a known negative prognostic factor, to predict mortality after DFU [32]. By constructing DFU, amputation and mortality risk maps on small scales, from a few hundred to a few thousand people, these studies offer a public health analysis that can facilitate actions of targeted prevention and care organization within a territory. They also make it possible to monitor public policies over time through repeated analyses.

To draw conclusions about the impact of social deprivation on care outcomes, it is important not to rely solely on analyses that are based on urban centers with specific socioeconomic factors, as this can complicate generalization. These city-specific analyses can, however, be used to guide local health authority actions.

Overall, studies using complex social deprivation scores based on both the average per capita income and a large population (not only focused on a hospital center) tend to show a deleterious effect of the average social deprivation of the habitat area on the occurrence and prognosis of DFU [34] and amputation [35]. This was also shown for the mortality of the people with diabetes [36]. Cox et al., showed an effect they call pull-up/pull-down. When a neighborhood is surrounded by more affluent neighborhoods, it tends to be “pulled up” and vice versa for an affluent neighborhood within more precarious neighborhoods [37]. This applied to the incidence of diabetes, as well as DFU and amputations, in the Glasgow area, even using a complex social deprivation score [2].

These observations of the negative impact of a socially deprived location on DFU prognosis have not been universally reported. In the retrospective cohort of patients published by the Baltimore team, there was no relationship between the average income of the home neighborhood and the prognosis for healing and amputation risk of a patient with a DFU [38]. Nevertheless, the authors pointed out that because their hospital was located in a disadvantaged area, people living in wealthy neighborhoods only went to this hospital as a last resort and only in the most serious cases. This finding, which is not isolated [39], reinforces the need to broaden the basis of analysis by trying to obtain a panel of data as exhaustive as possible of DFU in the area of study, taking into account all hospitals and all DFU managed in city care [40].

It should be noted that, for the previously cited Tayside analysis [41], the quintile of neighborhoods was not associated with a worse prognosis after adjusting for several clinical factors such as diabetes complications [41]. The effect of the residential area on a pathology and the social impact of the pathology are not necessarily observed in the management of its complications in countries where care is provided in a generally egalitarian manner. The organization of care may not be able to prevent the onset of an illness, but it may still be efficient in managing complications regardless of the person’s social background. A negative impact of the living environment can thus be compensated for by care organization that is effective for the most affected populations.

### 4.2. National-Level Studies

In the United States, Margolis et al., collected data on Medicare patients through hospitals and hospital referral regions. They highlighted areas at high risk of lower limb amputation, primarily in Texas, Louisiana, Mississippi, Arkansas and Alabama, and areas at low risk, principally in California, New Mexico, Michigan and Florida. They compared the sociodemographic characteristics of these areas and found that the high-risk areas had older populations with a high prevalence of African-American people living with diabetes and, notably, a lower average socioeconomic level [42]. Similarly, in the UK, higher rates of amputation and revascularization were found in the most disadvantaged neighborhoods, even though, taken as a whole, the most disadvantaged neighborhoods did not always appear significantly linked to over-risk at the national level [43]. Last, there also seems to be a hierarchy in the rate of amputation according to the patient’s region of residence, with more major amputations in the most precarious regions [44].

A striking hierarchy of risk based on the social deprivation of the neighborhood of residence can be observed even in states or countries with a social protection system considered to be universal [34], with a greater impact of the deprivation of the neighborhood of residence on men compared to women in terms of DFU prognosis [45]. These observations reinforce the idea that the cost barrier of access to basic care and the cost of a lack of access to hospitalization are not the only criteria that play a role in the impact of social deprivation on the prognosis of DFU. Hsu et al., showed that in Taiwan’s universal healthcare system, the most disadvantaged people had fewer HbA1c measurements, LDL controls and retinographies. For a pathology like the diabetic foot, which is based on a balance in the diabetes and where prevention is at the heart of management policies, the impact can be direct [46].

Regarding the ethnic factor, very curiously, Nishino et al., performed a large study in 2015 that took into account all the districts defined by the British Index of Multiple Deprivation (IMD), and showed that ethnicity contributed to the risk of hospitalization for diabetes complications [47]. They found that the social deprivation gradient was the same as that usually found with IMDs and that people whose ethnicity statistically corresponded to a population with an immigrant background had a higher risk of hospitalization, although they found no social deprivation gradient for these people, unlike for Caucasians [48]. It is unknown whether this observation was also valid for African-Americans or other ethnic groups at risk of infection or amputation [49] or post-amputation mortality [50], as has been suggested for end-stage renal disease [51]. These findings open future perspectives for exploration, especially since the observations were not systematically made in the British population [43].

## 5. Relationship between Healthcare Access Markers and Both Risk and Prognosis of DFU

The impact of access to a general practitioner seems to be important, even more than the access to a hospital [33]. Close monitoring and knowledge of the patient are crucial for the complex management of a patient’s condition during chronic disease, especially when an alteration in autonomy occurs. The hospital plays an essential role in care, of course, but focuses on technical and multidisciplinary care. In addition, although the notion of distance has an impact on survival for pathologies like stroke [52], it has much less of an impact on the healthcare system for pathologies that do not present as an immediate vital emergency, such as DFU in most cases. Yet, once again, the scale of the geographical area under study appears to change the conclusion of the observations. The previously cited study on the Scottish community of Tayside-Dundee showed no effect of distance (in time) from the general practitioner or hospital center on the risk of DFU and amputation [41]. The average time for access to a general practitioner was 6.48 min (SD: 5–25). For these very short times, it is indeed conceivable that there would be no difference in prognosis, but no consideration was given to the ability of the general practitioner to see the patient within a reasonable period. For example, a physician may be located across the street but not have emergency slots for several days. For another major complication of diabetes, diabetic nephropathy, the link with the distance from the professional seems more relevant, at least in the United States [53]. Furthermore, in a recent study, we highlighted the link between poor prognosis of DFU and poor access to a nurse in a southern region of France [54].

Wrobel et al., analyzed major amputation rates among Medicare patients by hospital and examined the impact of service strategies and “schools” of management on crude amputation rates and amputation ratios (major and minor) [55]. Care for diabetic foot is difficult, failure is common, and management strategies likely play a role. The access to trained, experienced and competent teams may also have a significant impact, with access to revascularization techniques being of prime importance [56]. Since the access to centers of excellence in revascularization is often limited to major urban centers, it would likely be more informative to distinguish between urban and rural populations than between favored and precarious populations in comparisons of the performance of DFU revascularization management.

Moreover, do the most socially deprived people have the same needs in terms of healthcare density? Beckles et al., also showed that people living with diabetes in the United States visited their diabetes specialist at least once a year whether or not they had health insurance but that those without insurance had fewer inspections of their feet [26]. This can probably be explained by a consultation directed toward other health problems, leaving no time for the control of the foot complications of diabetes in these people. Similarly, a British study showed that general practitioners had fewer chiropodists in their care center or network when they were located in disadvantaged areas [57]. This should be put into perspective with a previous study showing that, overall, higher socio-educational levels were associated with better efforts to prevent diabetes complications from the medical profession [58]. Furthermore, as noted, socioeconomic level and socio-educational level are independent factors that predict the delay in addressing DFU in specialized hospital centers [59].

## 6. Conclusions

In conclusion, at the individual level, the link between financial deprivation and both DFU risk and prognosis is relatively constant, even in subjects with universal health insurance. However, the correlation of both DFU risk and prognosis and social deprivation—as well as the strength and characteristics of this correlation—strongly depends on both the way social deprivation is calculated and the way in which questions about the social deprivation−DFU relationship are framed. At the neighborhood level, we come to similar conclusions. The social deprivation of the neighborhood can take precedence over the individual social level for impact on both DFU risk and prognosis. In terms of access to care, actual accessibility rather than distance also seems to play a role in both DFU risk and prognosis. Some neighborhoods with social precariousness may require a denser network of primary care and expert centers. There is, thus, a relationship between the risk and prognosis of DFU and the three factors of deprivation measured at the individual level, social deprivation measured at the neighborhood level, and access to care. However, it should be kept in mind that these factors may modulate each other.

We strongly encourage evaluation of public health policies in terms of the impact of the person’s social category on the risk and prognosis of DFU. The most socially deprived populations may warrant specific care programs that will need to be evaluated. Furthermore, it will be important to include subgroup analysis of social deprivation in studies that assess DFU risk and prognosis.

Finally, whatever the relationship between social deprivation and the diabetic foot, practitioners need to take into account each patient’s individuality, whether the patient is seen in consultation or is hospitalized. It is important always to keep in mind that individual characteristics and comorbidities still have the greatest impact on the prognosis and will determine treatment decisions [40,59,60].

## Data Availability

Not applicable.

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
