# Peer review of "Social Deprivation, Healthcare Access and Diabetic Foot Ulcer: A Narrative Review"

_jcm, 2022, doi:10.3390/jcm11185431_

Round 1
Reviewer 1 Report
Generally, the paper is relevant but I do have some comments about the way that it has been written in order to make the paper more understandable and easier to read. These are found below but is also attached.
1. Introduction
a. The investigators have failed to define from the beginning what is meant by social deprivation- a construct that is comprised of several other components. Which is why it is difficult to understand this paper. In the Introduction, this concept should probably be introduced at least how it is defined and measured, since socio-economic capability or profile is only one aspect. Based on the definition below, does the author just want to discuss social deprivation or both social deprivation and exclusion.
“The concepts of social deprivation and social exclusion share a similar focus on the inability of individuals to participate fully in the life of their community or society. The measurement of social deprivation has tended to emphasize a lack of material or financial resources that contributes to a lack of social participation, whereas measures of social exclusion have emphasized the lack of participation to a broader range of social, cultural, and political activities (Levitas et al., 2007).”
Tarani Chandola, Richard Conibere, in International Encyclopedia of the Social & Behavioral Sciences (Second Edition), 2015
From the American psychological Association: 1. limited access to society’s resources due to poverty, discrimination, or other disadvantage. See cultural deprivation. 2. lack of adequate opportunity for social experience.
b. Also, there are already many studies on the links between complications and social deprivation. What is special with DFU that it has to be separately studied from the other complications is also not clear from the Introduction/background. What was pointed out in the first paragraph is that amputations indicate the quality of the healthcare system, but what is the hypothesis about the association between social deprivation as defined above and the occurrence of DFU?
c. Not all DFU’s are actually “lower limb trophic disoders” so the 2 cannot be used interchangeably. This is the reason why the terminology of trophic disorders became obsolete – because it is not encompassing of all types of DFU.
d. It would have helped significantly if the review had some well-defined objectives at the onset so that we could anticipate what would be the content of the article. It was not clear to me whether the author/s were trying to establish the association between the occurrence of DFU and social deprivation, or the prognosis/outcomes of DFU and social deprivation. In some paragraphs, it was the former and in some the latter, or is it both?
2. Language and writing style
a. Improve language and writing style. There are also many typographicaly errors e.g. on page 2, paragraph 2, sentence 1: I”t is interesting to note that in two American states, Weissman et al. found higher 65 rates [2.27 (IC 95% 1.57-2.98) and 1.53 (IC95% 1.18-1.89) relative risk] of hospitalization for 66gangrene (diabetic and non-diabetic) for those without insurance or with Medicaid insur-ance than for those with private insurance. “ I think that should CI or confidence interval and NOT IC.
b. An example of inappropriate use of words is on page 2, paragraph 3, sentence 2: “It gives a score that takes into account not only income but also the social deprivation that individuals might experience, particularly regarding their ability to call on their entourage to help them. “ An entourage is defined as, “a group of people attending or surrounding an important person”. So. Maybe what is meant by the author is “caregivers” or “family” or “carers”.
c. Please also use person-centered language e.g. “diabetic patient” or “diabetic people” are no longer used but rather we now use “persons with diabetes”.
3. Materials and methods
a. While this is a narrative review, there should still be a description of the methodology by which the literature was searched to arrive at the decisions for those publications which were included in the review. It would be good for the authors to take a look at this quality assessment before re-submitting this paper: “SANRA, the Scale for the Assessment of Narrative Review Articles, a brief critical appraisal tool for the assessment of non-systematic articles.” Baethge C, Goldbeck-Wood S, Mertens S. SANRA-a scale for the quality assessment of narrative review articles. Res Integr Peer Rev. 2019 Mar 26;4:5. doi: 10.1186/s41073-019-0064-8 . PMID: 30962953; PMCID: PMC6434870
.
4. Discussion
a. Perhaps, there should first be a short discussion of what in fact comprises social deprivation so that we can then create a conceptual framework of what elements or factors will then be analyzed in terms of the risk and the prognosis for DFU.
5. Conclusion. Since there were no objectives that were given at the outset it hard to evaluate whether the conclusions are valid or whether the study intent has been reached, but the conclusions should be at least based on the major points in the outline and thus, what are the opportunities for further research.
A. Is there a link or association between Individual socioeconomic deprivation and diabetic foot ulcer risk and prognosis? If yes, is it for both financial deprivation and socioeducational factors?
B. What is the link between social deprivation of the area of residence and diabetic foot ulcer risk and prognosis? Does distance from the hospital matter? Or is the type of neighborhood a major factor?
C. Poor Healthcare access may not only be because lack of physical access to hospitals or to GP’s, but also because of lack of a medical insurance. How does this contribute to the incidence and the prognosis of persons with DFU?
Author Response
Dear Reviewer,
We thank you for your careful review which allows us to improve the manuscript. Please find our responses below.
Generally, the paper is relevant but I do have some comments about the way that it has been written in order to make the paper more understandable and easier to read. These are found below but is also attached.
- Introduction
- The investigators have failed to define from the beginning what is meant by social deprivation- a construct that is comprised of several other components. Which is why it is difficult to understand this paper. In the Introduction, this concept should probably be introduced at least how it is defined and measured, since socio-economic capability or profile is only one aspect. Based on the definition below, does the author just want to discuss social deprivation or both social deprivation and exclusion.
“The concepts of social deprivation and social exclusion share a similar focus on the inability of individuals to participate fully in the life of their community or society. The measurement of social deprivation has tended to emphasize a lack of material or financial resources that contributes to a lack of social participation, whereas measures of social exclusion have emphasized the lack of participation to a broader range of social, cultural, and political activities (Levitas et al., 2007).”
Tarani Chandola, Richard Conibere, in International Encyclopedia of the Social & Behavioral Sciences (Second Edition), 2015
From the American psychological Association:
1. limited access to society’s resources due to poverty, discrimination, or other disadvantage. See cultural deprivation. 2. lack of adequate opportunity for social experience.
Response: We thank you for this remark which allows us to give definition of terms used. There is indeed a distinction between financial problems and social exclusion. However, literature is not consistent in its definition of social deprivation. The literature on the diabetic foot ulcer (DFU) uses the term "social deprivation" in a broad sense, which leads us to differentiate between strict financial deprivation and social deprivation in the same broad sense, including social exclusion. We have chosen to make a distinction between:
1- Financial deprivation
2- Social deprivation/social exclusion in the individual sense
3- Deprivation of the place of living because it is often studied separately.
This has been included in the text
- Also, there are already many studies on the links between complications and social deprivation. What is special with DFU that it has to be separately studied from the other complications is also not clear from the Introduction/background. What was pointed out in the first paragraph is that amputations indicate the quality of the healthcare system, but what is the hypothesis about the association between social deprivation as defined above and the occurrence of DFU?
Response: We also thank and hope that the details provided will clarify the specificity of diabetic foot ulcers.
Its specificity lies mainly in the high consumption of hospital and ambulatory care, and in particular in human needs. There is also a very strong social impact with prolonged bed rest. This, compared to other complications of diabetes.
- Not all DFU’s are actually “lower limb trophic disoders” so the 2 cannot be used interchangeably. This is the reason why the terminology of trophic disorders became obsolete – because it is not encompassing of all types of DFU.
Response: Unfortunately, we were not able to take this point fully into account because epidemiological studies do not always make distinction.
Data on the prevalence or incidence of DFU are missing in many databases. It is why, it is often necessary to focus on lower limb amputations with the attendant risks of bias (not including only DFU). This is justified by the fact that diabetes is the leading cause of non-traumatic lower limb amputation in developed countries.
We systematically differentiated between the risk of DFU incidence (when the database studied offers this data) and the risk of amputation/mortality (DFU prognosis) when this is the only data available. Time to healing is unfortunately a data that is difficult to evaluate, which makes it often absent from studies.
- It would have helped significantly if the review had some well-defined objectives at the onset so that we could anticipate what would be the content of the article. It was not clear to me whether the author/s were trying to establish the association between the occurrence of DFU and social deprivation, or the prognosis/outcomes of DFU and social deprivation. In some paragraphs, it was the former and in some the latter, or is it both?
Response: We have tried to make the objectives clearer. Our ends are to answer to the following questions:
- Is there a relationship between individual-level social deprivation markers on both risk and prognosis of DFU.
- Is there a relationship between neighbourhoods social deprivation markers on both risk and prognosis of DFU.
- Is there a relationship between healthcare access markers on both risk and prognosis of DFU.
- Language and writing style
- Improve language and writing style. There are also many typographicaly errors e.g. on page 2, paragraph 2, sentence 1: I”t is interesting to note that in two American states, Weissman et al. found higher 65 rates [2.27 (IC 95% 1.57-2.98)and 1.53 (IC95% 1.18-1.89) relative risk] of hospitalization for 66gangrene (diabetic and non-diabetic) for those without insurance or with Medicaid insur-ance than for those with private insurance. “ I think that should CI or confidence interval and NOT IC.
Response: We thank you for your vigilance. We have corrected the items mentioned.
- An example of inappropriate use of words is on page 2, paragraph 3, sentence 2: “It gives a score that takes into account not only income but also the social deprivation that individuals might experience, particularly regarding their ability to call on their entourageto help them. “ An entourage is defined as, “a group of people attending or surrounding an important person”. So. Maybe what is meant by the author is “caregivers” or “family” or “carers”.
Response: We thank you for this remark. The change was indeed significant.
- Please also use person-centered language e.g. “diabetic patient” or “diabetic people” are no longer used but rather we now use “persons with diabetes”.
Response: This is indeed important. The changes are made.
- Materials and methods
- While this is a narrative review, there should still be a description of the methodology by which the literature was searched to arrive at the decisions for those publications which were included in the review. It would be good for the authors to take a look at this quality assessment before re-submitting this paper: “SANRA, the Scale for the Assessment of Narrative Review Articles, a brief critical appraisal tool for the assessment of non-systematic articles.” Baethge C, Goldbeck-Wood S, Mertens S. SANRA-a scale for the quality assessment of narrative review articles. Res Integr Peer Rev. 2019 Mar 26;4:5. doi: 10.1186/s41073-019-0064-8 . PMID: 30962953; PMCID: PMC6434870
.
Response: We thank you for this clarification. We have added a method section.
We followed the evaluation recommendations of Baethge et al.
- Discussion
- Perhaps, there should first be a short discussion of what in fact comprises social deprivation so that we can then create a conceptual framework of what elements or factors will then be analyzed in terms of the risk and the prognosis for DFU.
Response: We thank you for this remark which improves the readability of the review. We have included our three levels of analysis in the conclusion
- Conclusion. Since there were no objectives that were given at the outset it hard to evaluate whether the conclusions are valid or whether the study intent has been reached, but the conclusions should be at least based on the major points in the outline and thus, what are the opportunities for further research.
- Is there a link or association between Individual socioeconomic deprivation and diabetic foot ulcer risk and prognosis? If yes, is it for both financial deprivation and socioeducational factors?
- What is the link between social deprivation of the area of residence and diabetic foot ulcer risk and prognosis? Does distance from the hospital matter? Or is the type of neighborhood a major factor?
- Poor Healthcare access may not only be because lack of physical access to hospitals or to GP’s, but also because of lack of a medical insurance. How does this contribute to the incidence and the prognosis of persons with DFU?
Response: We thank you for this suggestion and have tried to be more precise in our conclusion by using the results of the review.
Reviewer 2 Report
Narrative review on a very relevant problem, namely the effect of social deprivation and healthcare access on diabetic foot ulcer (DFU). The review is however not well structured, there is a lack of clear definitions, prevention of DFU and treatment of DFU are not separated, and it is not clear what the conclusions are.
I miss a methods section.
- How did the authors select their papers?
- What aspects of social deprivation will be looked at: income, employment, education level, ethnic origin, …
- What aspects of health care access and quality will be looked at: availability of 1° care, of foot clinics, time between onset of the problem and presentation for care, time between 1st visit to 1°care and referral to a multidisciplinary diabetic foot clinic, …
- What aspect of DFU will they take into account: occurrence, time to healing, minor amputation, major amputation, mortality, …
This would help in structuring the paper
What are the conclusions? The actual conclusion section is written as a new introduction, it is no conclusion of the review.
The authors look at the extra costs for patients and mention footware. There are however a lot of additional extra costs possible for DFU patients, such as inadequate reimbursement of wound care products, of vacuum therapy, costs of transport, …
Author Response
Dear Reviewer,
We thank you for your careful review which allows us to improve the manuscript. Please find our responses below.
Narrative review on a very relevant problem, namely the effect of social deprivation and healthcare access on diabetic foot ulcer (DFU). The review is however not well structured, there is a lack of clear definitions, prevention of DFU and treatment of DFU are not separated, and it is not clear what the conclusions are.
Response: We have tried to clarify the subject of analysis and the conclusions.
Data on the prevalence or incidence of DFU are missing in many databases. It is why, it is often necessary to focus on lower limb amputations with the attendant risks of bias (not including only DFU). This is justified by the fact that diabetes is the leading cause of non-traumatic lower limb amputation in developed countries.
We systematically differentiated between the risk of DFU incidence (when the database studied offers this data) and the risk of amputation/mortality (DFU prognosis) when this is the only data available. Time to healing is unfortunately a data that is difficult to evaluate, which makes it often absent from studies.
I miss a methods section.
Response: You are absolutely right and we apologize for this. The methods section has been added.
- How did the authors select their papers?
Response: We have included this important point.
- What aspects of social deprivation will be looked at: income, employment, education level, ethnic origin, …
Response: You are right, there is indeed a distinction between financial problems and social exclusion. However, literature is not consistent in its definition of social deprivation. The literature on the diabetic foot ulcer (DFU) uses the term "social deprivation" in a broad sense, which leads us to differentiate between strict financial deprivation and social deprivation in the same broad sense, including social exclusion. We have chosen to make a distinction between:
1- Financial deprivation
2- Social deprivation/social exclusion in the individual sense
3- Deprivation of the place of living because it is often studied separately.
We have specified as much as possible for each study the specific aspects studied: education, financial, ethnics.
For example, the level of education does not have the same effect on women as on men.
We have also focused studies using complex scales (SIMD for instance) that take into account several types of social deprivation. The scales are systematically cited in the bibliography.
- What aspects of health care access and quality will be looked at: availability of 1° care, of foot clinics, time between onset of the problem and presentation for care, time between 1st visit to 1°care and referral to a multidisciplinary diabetic foot clinic, …
Response: This point is very important, we agree. However, this question is quite difficult to answer as it is not consensual in clinical studies. The great diversity of the main judgment criteria taken into account in the various studies often make them not very comparable. We therefore had to comment on the available data and try to assemble them into a coherent structure.
We focused on two types of access to care: access to primary care (general practitioner and nurse) and access to a specialized center.
In both cases, the notions of distance (km or min) or real accessibility (number of patients per practitioner for example) are analyzed. This is systematically specified and commented on.
- What aspect of DFU will they take into account: occurrence, time to healing, minor amputation, major amputation, mortality, …
Response: Data on the prevalence or incidence of DFU are missing in many databases. It is why, it is often necessary to focus on lower limb amputations with the attendant risks of bias (not including only DFU). This is justified by the fact that diabetes is the leading cause of non-traumatic lower limb amputation in developed countries.
We systematically differentiated between the risk of DFU incidence (when the database studied offers this data) and the risk of amputation/mortality (DFU prognosis) when this is the only data available. Time to healing is unfortunately a data that is difficult to evaluate, which makes it often absent from studies.
It has been specified in the introduction.
This would help in structuring the paper
What are the conclusions? The actual conclusion section is written as a new introduction, it is no conclusion of the review.
Response: We have attempted to clarify the findings using our three levels of analysis:
- Is there a relationship between individual-level social deprivation markers on both risk and prognosis of DFU.
- Is there a relationship between neighbourhoods social deprivation markers on both risk and prognosis of DFU.
- Is there a relationship between healthcare access markers on both risk and prognosis of DFU.
The authors look at the extra costs for patients and mention footware. There are however a lot of additional extra costs possible for DFU patients, such as inadequate reimbursement of wound care products, of vacuum therapy, costs of transport, …
Response: This element is an example to illustrate the impact of hidden health costs in a universal system. However, we have added your very valid point.
Round 2
Reviewer 1 Report
Thank you for revising the paper according to most of the comments and recommendations. It now has greater clarity and better readability, and the results and conclusions are better written and well emphasized.
I have no further comments. Congratulations!
Author Response
Reviewer 1
Open Review
( ) I would not like to sign my review report
(x) I would like to sign my review report
English language and style
( ) Extensive editing of English language and style required
( ) Moderate English changes required
(x) English language and style are fine/minor spell check required
( ) I don't feel qualified to judge about the English language and style
Comments and Suggestions for Authors
Thank you for revising the paper according to most of the comments and recommendations. It now has greater clarity and better readability, and the results and conclusions are better written and well emphasized.
I have no further comments. Congratulations!
We thank you again for your reviewing, that helped a lot in improving the review. For this last revision, proofreading was done by a native English-speaking.

Reviewer 2 Report
I thank the authors for their answers to my questions and suggestions.
A methods section was added and the conclusions were formulated more sharply.
I still find the paper difficult to read, partly because it is difficult to structure this topic much, partly because the authors are not native speakers of English.
Author Response
Reviewer 2
Open Review
(x) I would not like to sign my review report
( ) I would like to sign my review report
English language and style
(x) Extensive editing of English language and style required
( ) Moderate English changes required
( ) English language and style are fine/minor spell check required
( ) I don't feel qualified to judge about the English language and style
Comments and Suggestions for Authors
I thank the authors for their answers to my questions and suggestions.
A methods section was added and the conclusions were formulated more sharply.
I still find the paper difficult to read, partly because it is difficult to structure this topic much, partly because the authors are not native speakers of English.
We thank you again for your reviewing, that helped a lot in improving the review. For this last revision, proofreading was done by a native English-speaking.